# The Effect of Persistence of Physical Exercise on the Positive Psychological Emotions of Primary School Students under the STEAM Education Concept

**DOI:** 10.3390/ijerph191811451

**Published:** 2022-09-12

**Authors:** Yubin Yuan, Xueyan Ji, Xiaoming Yang, Chen Wang, Shamsulariffin Samsudin, Roxana Dev Omar Dev

**Affiliations:** 1Department of Sports Studies, Faculty of Educational Studies, Universiti Putra Malaysia, Serdang 43400, Selangor, Malaysia; 2College of Physical Education, East China University of Technology, Nanchang 330013, China

**Keywords:** STEAM education concept, physical exercise, positive emotion, mental health, correlation, regression

## Abstract

The effect of persistence of physical exercise on the psychological and emotional aspects of primary school students is studied to improve the comprehensive quality of current Chinese primary school students and explore the effect of physical exercise on students’ emotions under the science, technology, engineering, art, mathematics (STEAM) education concept. First, students in a primary school in Nanchang are taken as the survey participants. Second, by formulating a physical exercise scale and a psychological and emotional scale, the current situation of physical exercise of primary school students is investigated by means of mathematical statistics. Finally, the current situation of physical exercise and the overall situation of positive psychological emotions of primary school students are analyzed, and the effect of physical exercise on the positive psychological emotions of primary school students is studied. The data show that there are significant differences in the amount of exercise and its three dimensions of intensity, time, and frequency, as well as the scores of positive emotions in the gender dimension, with the boys scoring higher than the girls. In terms of grades, students in grades 1, 2, and 6 are higher than students in grades 3, 4, and 5 on the level of a small amount of exercise, while students in grades 3, 4, and 5 are higher than the other three grades in terms of a moderate amount of exercise. Moreover, in the aspect of positive psychological emotions, the lower-grade students are obviously higher than the upper-grade students, and the second- and third-grade students present marginal significance, *p* = 0.058. The correlation and regression between physical exercise and positive psychological emotions are calculated and analyzed, and it is found that there is a significant positive correlation between physical exercise indicators and positive psychological emotions, with a correlation coefficient of 0.297. Physical exercise explains 8.8% of positive emotions. This research also makes relevant recommendations for students and schools and has played a role in strengthening the physical exercise and mental health of primary and secondary school students. Greater attention to the physical exercise of primary school students is recommended.

## 1. Introduction

With the background of quality education as the pillar of the future development of the country, the development of positive emotions is very important for primary school students. The current educational background also puts forward higher requirements for the comprehensive quality of primary school students [1]. Quality education at this stage should not only pay attention to the development of students’ academic studies but also focus on the improvement in all-around comprehensive quality, such as students’ physical quality and psychological quality. Persistence in physical exercise can enhance students’ physiques and hone their will, which is of great significance for students’ comprehensive development. This research plays a vital role in the study of students’ comprehensive quality under the scientific–educational concept of the relationship and role between students’ physical exercise and positive emotions [2].

Adhering to physical exercise can not only improve the physical quality of students but also advance their psychological quality. All schools should strengthen their understanding of the persistence of physical exercise and attach importance to physical development. To improve the comprehensive quality of the current domestic primary school students, under the concept of science, technology, engineering, art, mathematics (STEAM) education, the impact of persistent physical activity on the psychological and emotional impact of primary school students has been explored. At present, there are many studies on the relationship between physical exercise and psychological emotions. Many studies have shown that physical exercise has an impact on psychological emotions. Klaperski and Fuchs (2021) [3] studied the physical exercise patterns of 52 anxious and depressed individuals and found that walking and jogging, two types of physical activity, significantly reduced levels of excessive anxiety and depression. Zhu et al. (2020) [4] measured participants after physical exercise and indicated that their levels of state anxiety, depression, tension, and psychological disturbance were significantly reduced, and their energy and pleasure levels were evidently improved. Rahawi et al. (2021) [5] found that physical exercise can effectively improve the ability of emotion regulation, maintain the positive emotions of college students, and promote the physical and mental health of students. To expand and construct a theory that emphasizes the importance of positive emotions, Holzer et al. (2021) [6] constructed a positive emotion scale based on this theory to examine the mediating role of resilience between hope and subjective well-being. Most of the current research is aimed at adults, mainly to study the psychological and physiological effects of physical exercise on adults, while there are few studies on the psychological and emotional effects of the persistence of physical exercise on primary school students. The educational concept of STEAM focuses on practice, especially for sports disciplines. STEAM education integrates the characteristics of various disciplines and is a fusion and creation of diverse disciplines.

To explore the impact of the persistence of physical exercise on the psychological and emotional aspects of primary school students under the STEAM education concept, first, the STEAM education concept and positive emotions are introduced. Second, by formulating the scale and using mathematical statistics, the current status of physical exercise of primary school students is investigated. Through this analysis and the overall situation of positive psychological emotions, the influence of physical exercise on the positive psychological emotions of primary school students is obtained. Finally, the results are discussed and relevant suggestions are made. This research aims to provide a positive effect on promoting the comprehensive development of primary and secondary school students.

## 2. Basic Theory and Research Methods

### 2.1. STEAM Education Concept

STEAM is an acronym for five words: science, technology, engineering, arts, and mathematics. STEAM education evolved from the STEM education plan and is a comprehensive education that integrates science, technology, engineering, mathematics, humanities, and arts [7]. Under the STEAM education concept, relying on the core literacy of physical education and pointing to the practical exploration of “interdisciplinary integration” in physical education teaching, it can further expand students’ learning horizons and form a multi-disciplinary and dynamic education. It is of great significance to promote the deep learning of students and promote the cultivation of compound talents [8].

STEAM education is a new learning concept. It changes the traditional teaching model and is closely linked to current education [9]. The method is closely related to physical education based on the STEAM education concept. It is in line with the research direction of China’s teaching reform to focus on improving the teaching mode and integration of teaching objectives through various thinking modes. These reforms aim to strengthen the attractiveness of physical exercise to students and to promote students’ self-motivation [10].

The purpose of STEAM education is to let students learn to create more learning styles suitable for real life, such as learning to listen to the opinions of others and trying a variety of ideas to solve problems. From this, students’ technical literacy and humanistic and artistic literacy in physical education should be valued and combined with STEAM education to cultivate students’ psychological characteristics of health, safety, self-confidence, and innovation [11].

### 2.2. Positive Emotion Theory

Many researchers have made specific descriptions or definitions for positive emotions. Moss et al. once proposed that positive emotions are good feelings when things are going well [12]. Ren et al. believed that positive emotion was an immediate response to something meaningful to the individual [13]. Her et al. considered that positive emotion was the feelings that arise when making progress in the process of achieving goals or being positively evaluated by others [14].

Positive emotions can have an uplifting effect on the organism. Positive emotions can add new power to people’s nervous system, can fully realize the potential of the organism, and improve the efficiency and endurance of mental and physical work [15,16]. Positive emotions are often stimulated by a sense of responsibility, professionalism, expectations, goals, and a sense of honor. Therefore, the way to maintain positive emotions is to make oneself have a sense of responsibility, honor, and professionalism as soon as possible, have short-term and long-term goals, and unremittingly strive and struggle to achieve the set goals [17]. Studies have demonstrated that positive emotions can increase adrenaline in the blood, the hormone that mobilizes the power of the organism, thereby making the person more empowered to achieve their goals [18,19]. Positive emotions are an important condition and marker for maintaining mental health [20].

### 2.3. Survey Participants and Research Methods

This research used primary school students as participants, took the relationship between physical exercise and positive psychological emotions as the research content, and randomly selected 300 primary school students from Nanchang primary schools as participants by random sampling method. The standard is selected from the “Physical Activity Rating Scale (PARS-3)” and positive emotion table. The “Physical Activity Rating Scale (PARS-3)” was compiled by Japanese researcher Koshimoto Hashimoto and then revised by researchers, such as Liang Deqing from the Wuhan Institute of Physical Education. The scale measures the amount of exercise of physical exercisers in three parts: the intensity of physical activity, the time of each exercise, and the frequency of exercise, and has high reliability and validity [21,22]. The scoring method is written as an Equation (1):“Amount of exercise” = “Intensity of exercise” × “Time of exercise” × “Frequency of exercise”(1)

Zhang et al. (2020) [23] point out exercise intensity and exercise frequency are divided into 1–5 cases, with 1–5 points, respectively, and exercise time is split into 1–5 types, with 0–4 points, respectively. According to the question options, in the aspect of exercise intensity, it is split into five intensities from A to E. In terms of exercise frequency, it is divided into five frequencies from A to E. In terms of time, it is also divided into five types from A to E according to the options. The highest score for the amount of exercise is 100 points, and the lowest score is 0 points. The standard of the amount of exercise is: a small amount of exercise is less than or equal to 19 points, a medium amount of exercise is between 20 and 42 points, and a large amount of exercise is 43 points or more [24]. The retest reliability of the scale is 0.82. The specific content of the scale is exhibited in Table 1:

Regarding the content of the questionnaire, considering that the age of the students cannot be answered in detail and accurately, the questionnaire is distributed to the students. With the help of their parents, answered by the students, parents assist to fill out. 

The measurement of personality characteristics is more suitable in the scope of the survey of psychological resilience. According to the actual situation in China, the positive and negative emotion scale compiled by Watson et al. was revised again, and positive emotions were selected to form a positive emotion scale [25].

The reliability and validity of the positive emotion scale were tested by relevant experts and teachers. After reviewing the contents of the scale, the results met the requirements and the scale had certain reliability. Further reliability test was carried out using Cronbach’s α coefficient [26]. Through repeated testing, the internal consistency of the α coefficient is 0.821, which indicates good reliability.

In this research, the content of positive emotions generally includes: interested, excited, strong, enthusiastic, proud, sensitive, and determined [27]. It is divided into 5 situations from “never” to “always”, with 0–4 points, respectively. The higher the score, the higher the individual’s level of positive emotion [28]. A mean of more than 2 is considered positive and good [29]. The specific scale is displayed in Table 2:

Regarding the positive emotion scale, in terms of understanding, students of diverse grades will affect the filling of the scale due to the difference in intelligence. Parents are required to assist children in completing the scale. Parents do auxiliary work and explain to students. Answered by the students, parents assist to fill out. The scale is an overall scale, and different variable interpretations will be carried out for students of different grades so that students of each grade can successfully complete the survey task.

Excel is used to input and process the basic situation of physical exercise and positive emotions of primary school students, and the details are discussed. Statistical software Statistical Product and Service Solutions (SPSS 19.0, Los Angeles, CA, USA) 25.0 is employed to process and analyze the data. Regression analyses are performed on variables related to physical exercise and positive emotions [30,31].

## 3. Results and Analysis

### 3.1. Description of the Basic Situation of Primary School Students Participating in Physical Exercise

In this survey, a total of 300 primary school students were randomly selected from a primary school in Nanchang as participants. A total of 300 scales were distributed, and 289 were actually recovered, with a recovery rate of 96.33%, of which 280 were valid scales, with an effective rate of 96.89%. The specific composition of the sample is presented in Table 3:

In Table 3, among the investigated primary school students, the number of boys is 122, accounting for 43.57%, and the number of girls is 158, accounting for 56.43%. From the first grade to the sixth grade, the numbers are 53, 51, 50, 32, 53, and 41, respectively. The statistics of exercise intensity are exhibited in Figure 1:

Figure 1 denotes that there is a significant difference in the intensity of physical activity among primary school students in the three primary schools in terms of gender. After calculation, χ^2^ = “13.155”, *p* = “0.011 < 0.05”. Especially for the participants with moderate training intensity, boy participants are more than girl participants, accounting for 79.7% and 69.1%, respectively. In addition, the exercise intensity of students in the lower grades is relatively small. With the increase in grades, the exercise intensity of students is gradually increasing. 

The statistical results for exercise time are expressed in Figure 2:

Figure 2 indicates that there is an extremely significant difference between boys and girls in the exercise time of each physical exercise. After calculation, χ^2^ = “20.727” and *p* = “0.000 < 0.001”. Among the people who exercise for more than half an hour each time, the proportion of boys is often higher than that of girls; the proportions are 59.8% and 53.5%, respectively, indicating that boys generally spend more time on physical exercise than girls. It may be due to the influence of living habits and the traditional concept of parents; there are obviously different attitudes towards physical exercise for boys and girls. In the aspects of grade variables, there is also an obvious difference in the time of physical exercise in different grades, χ^2^ = “21.066” and *p* = “0.049 < 0.05”. In general, with the increase in grades, the time for physical exercise gradually decreases. The reason is that, as the grade increases, the proportion of students’ cultural courses gradually increases, which relatively takes up part of the students’ physical exercise time.

According to the third question in Table 1, the statistical results of exercise frequency are revealed in Figure 3:

In Figure 3, there is a great difference in the frequency of physical exercise between boys and girls. After calculation, χ^2^ = “43.980” and *p* = “0.000 < 0.001”. The proportion of girls’ exercise frequency less than three times a week is 60.7%, which is significantly higher than that of boys’ 43.8%. The proportion of boys who do physical exercise three to five times a week is 38.3%, while that of girls is 26.9%, illustrating that boys participate in physical exercise more frequently than girls. It may be influenced by the living habits and the traditional concept of parents; there are obviously different attitudes towards physical exercise for boys and girls. There is also a distinct difference in the frequency of physical exercise for students of different grades, χ^2^ = “33.319” and *p* = “0.001 < 0.01”. The percentage of the six grades is basically the same, with one to two times a week and three to five times a week, but the rate of once a day for the sixth-grade students is lower than that of other grades. This is mainly because the sixth grade is related to the entrance examination from elementary school to junior high school, so students have less time for sports activities.

The associated statistics for the amount of exercise are demonstrated in Figure 4:

Figure 4 shows that there is a significant difference in the level of the amount of exercise between boys and girls, χ^2^ = “13.155”, *p* = “0.000 < 0.001”. In the two levels of moderate amount of exercise and high amount of exercise, boys accounted for 58.9% and 23.8%, respectively, higher than girls’ 52.4% and 14.3%. The proportion of girls in the small amount of exercise is 33.3%, which is higher than that of boys, 17.9%. It reflects that boys exercise more than girls in sports. Students of different grades also have great differences in the amount of exercise, χ^2^ = “103.92”, *p* = “0.044 < 0.05”. Students in grades 1, 2, and 6 are higher than those in grades 3, 4, and 5 on the level of a small amount of exercise, while students in grades 3, 4, and 5 are higher than the other three grades in terms of a moderate amount of exercise.

### 3.2. Analysis of the Overall Situation of the Positive Psychological Emotions of Primary School Students

The average score for positive emotions of the samples is 2.15 points, with a standard deviation of 0.42. The positive emotion scale uses a five-point scale of 0–4, with a median of 2. The research results reveal that the average score of positive emotions is 2.15, which is higher than the average value, indicating that the level of positive emotions of primary schoolchildren in these three schools is higher, and they can experience more positive emotions in daily life. Figure 5 refers to the differences in positive psychological emotions for gender and grade.

In Figure 5, there are obvious differences in the scores of positive emotions in the gender dimension, with the boys scoring higher than the girls. The independent sample *t*-test is 4.073, *p* = “0.000 < 0.001”. As for the difference in grades, the lower-grade students are significantly higher than the upper-grade students in this aspect, and the second- and third-grade students are marginally significant in the aspect of emotional positivity, *p* = “0.058”.

### 3.3. Analysis of the Effect of Physical Exercise on Positive Psychological Emotions

Figure 6 and Table 4 illustrate the differences in positive emotions in physical exercise:

Figure 6 and Table 4 signify that there are significant differences in positive emotions in all dimensions of physical exercise. A further comparison found that there are great differences in the amount of exercise. In terms of exercise intensity, there is no distinct difference in positive psychological emotions in the two dimensions of low intensity and small intensity, and there are high differences in other intensities. In the aspect of duration, the scores of positive emotions are between “under 10 min” and “11 min to 20 min”. There are no significant differences between “31 min to 59 min” and “more than 60 min”. Further, the rest of the durations are significantly different. From the perspective of exercise frequency, the total positive emotion score is an obvious difference between “about once a day” and “about twice a day”, and between “1–2 times a week” and “3–5 times a week”.

The correlation analysis results of physical exercise and positive psychological emotions are illustrated in Table 5:

Table 5 indicates that there is a significant positive correlation between physical exercise indicators and positive psychological emotions. To detect whether physical exercise has an obvious predictive effect on positive emotions, positive emotion was used as the dependent variable and physical exercise as the predictor variable to establish a linear regression model to explore the causal relationship between fitness exercise and positive emotion. The result is as follows:

Table 6 indicates that physical exercise finally entered the regression model of positive emotions, with a correlation coefficient of 0.297, and physical exercise explained 8.8% of positive emotions.

Through the analysis of the relationship between physical exercise and positive emotions, it is found that there are significant differences between positive emotions and physical exercise and various dimensions. The best time for one exercise is more than 30 min, and there should be at least one physical exercise per week. There is an obvious difference in exercise intensity. Through correlation analysis, it is found that various dimensions of physical exercise are significantly positively correlated with positive emotions. Regression analysis also testifies that physical exercise has a positive guiding effect on positive emotions. In addition to advocating the importance of physical exercise, this research also aims to improve the mental health of primary and secondary school students. Therefore, it is recommended that schools attach importance to the practical application of STEAM education concepts. Schools are recommended to popularize mental health knowledge, grasp the relationship between physical education and psychological emotions, and to regard physical exercise as an important educational means to promote mental health. There are many factors affecting the development of mental health, and schools should take gender, grade, age, and other factors into consideration.

## 4. Conclusions

The purpose was to promote improvement in all-around comprehensive quality of primary education and strengthen the emphasis on physical exercise and mental health levels of primary school students. Under the STEAM education concept, the effect of physical exercise on the positive psychological emotions of primary school students was studied. First, the concept of STEAM education and positive psychological emotions were expounded. Second, 300 primary school students in a primary school in Nanchang were used as the survey participants to investigate the current status of physical exercise and the overall situation of positive psychological emotions of primary school students. Finally, through the analysis of the survey data, it was concluded that physical exercise had an effect on the positive psychological emotions of primary school students. The survey results testify that there are significant differences between boys and girls in the three dimensions of the amount of exercise and including exercise intensity, exercise time, and exercise frequency. In the two levels of moderate and high amounts of exercise, boys accounted for 58.9% and 23.8%, respectively, higher than girls’ 52.4% and 14.3%. The proportion of girls in the small amount of exercise is 33.3%, which is higher than that of boys, 17.9%, which reflects that boys exercise more than girls in sports. There are still obvious differences in the scores of positive emotions in the gender dimension, with the boys scoring higher than the girls. For different grades, there are also great differences in the amount of exercise. Students in grades 1, 2, and 6 are higher than those in grades 3, 4, and 5 on the level of a small amount of exercise, while students in grades 3, 4, and 5 are higher than the other three grades in terms of a moderate amount of exercise. Moreover, the positive psychological emotions of lower-grade students are higher than those of upper-grade students. Through the analysis of the correlation and regression between physical exercise and positive psychological emotions, it is concluded that there is a great positive correlation between physical exercise indicators and positive psychological emotions. Furthermore, this research also proves the effect of physical exercise on positive emotions. It is also recommended that schools enhance the experience of positive emotions: students should cheer up in the face of difficulties, change unfavorable conditions in reality, and improve their positive emotions in the face of setbacks. Let students experience happy emotions in the process of sports, continuously stimulate positive emotions, and attach importance to the practical application of STEAM education concepts. Due to limited energy, the influence of family conditions and other aspects are not considered for the differences in samples, and more detailed exploration will be added in the follow-up. This research had a positive effect on promoting the comprehensive development of primary school students.

## 5. Limitations

The limitation is that the study group is small, so the scope of the group that can refer to this research for improvement is limited to primary school students. In addition, there are few research methods used, and the content of the research on the sports status of primary school students is not detailed enough, so the research on the specific sports status of primary school students is not clear enough. To sum up, in future research, the scope of the research group should be expanded first, thereby improving the mechanism of the research. It is also necessary to adopt more detailed methods to comprehensively explore the specific exercise conditions of each group to provide a reference for the improvement of emotions of more groups.

## Figures and Tables

**Figure 1 ijerph-19-11451-f001:**
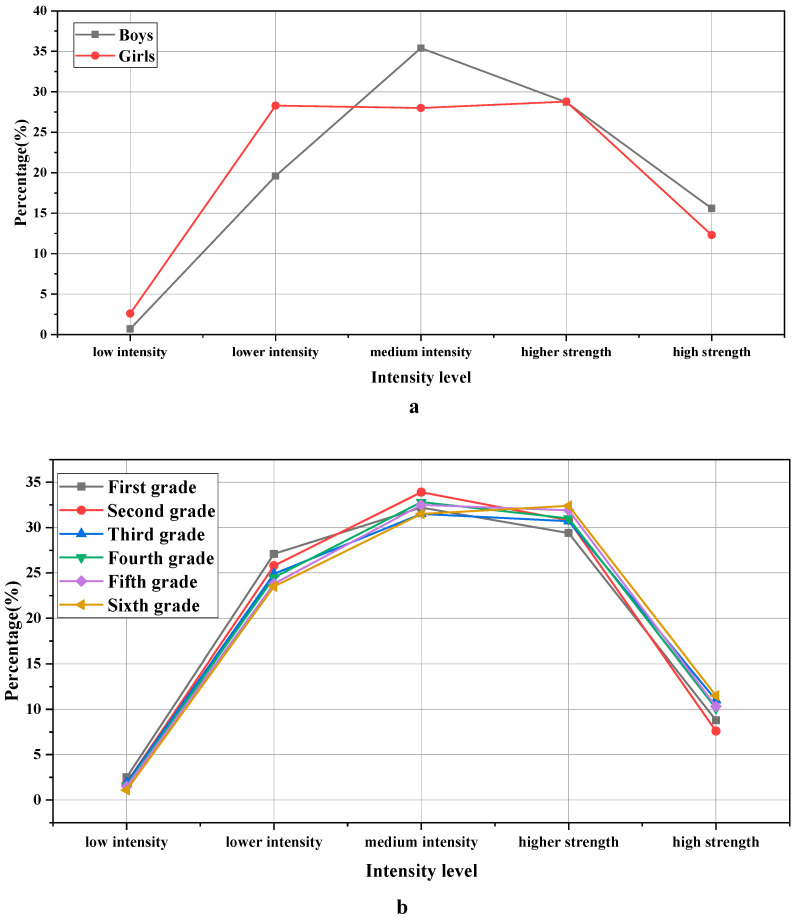
Statistical results of exercise intensity: (**a**) indicates the gender classification of the sample; (**b**) indicates the grade classification of the sample.

**Figure 2 ijerph-19-11451-f002:**
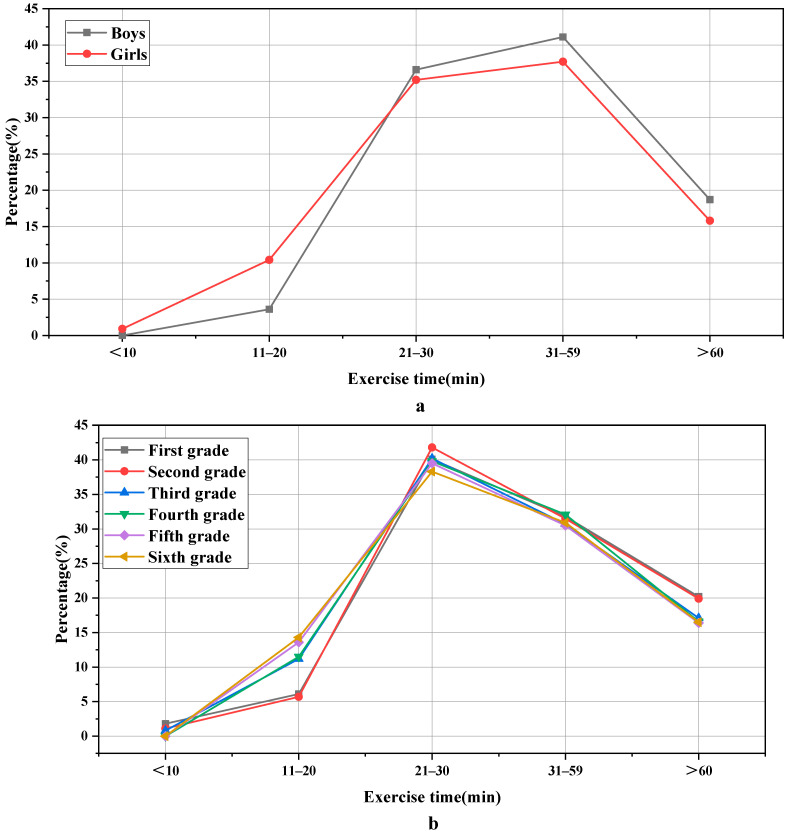
Statistical results of exercise time: (**a**) stands for the gender classification of the sample; (**b**) stands for the grade classification of the sample.

**Figure 3 ijerph-19-11451-f003:**
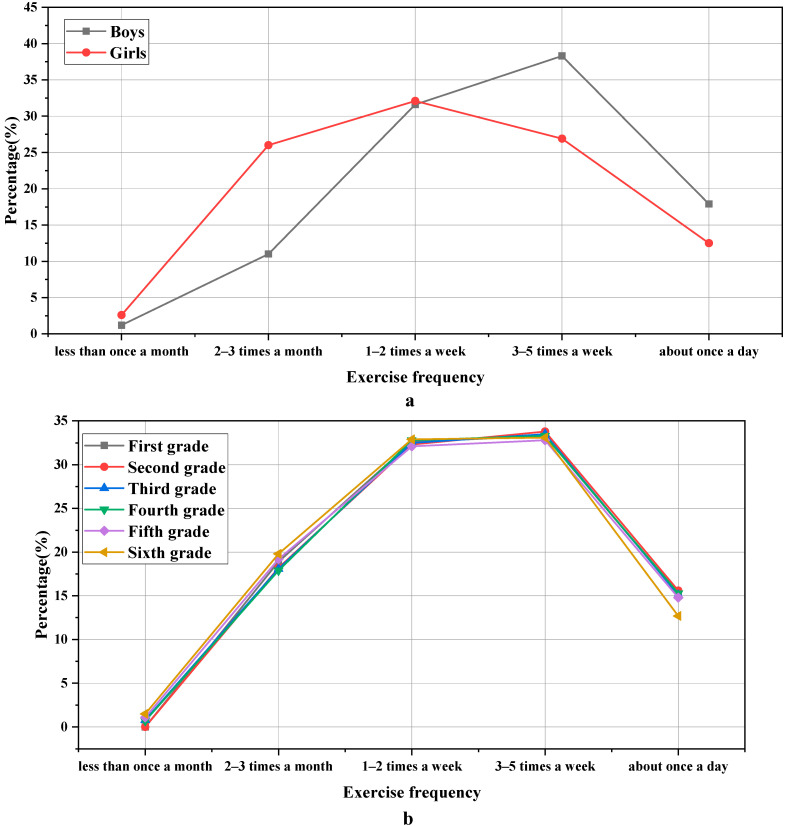
Statistical results of exercise frequency: (**a**) refers to the gender classification of the sample; (**b**) refers to the grade classification of the sample.

**Figure 4 ijerph-19-11451-f004:**
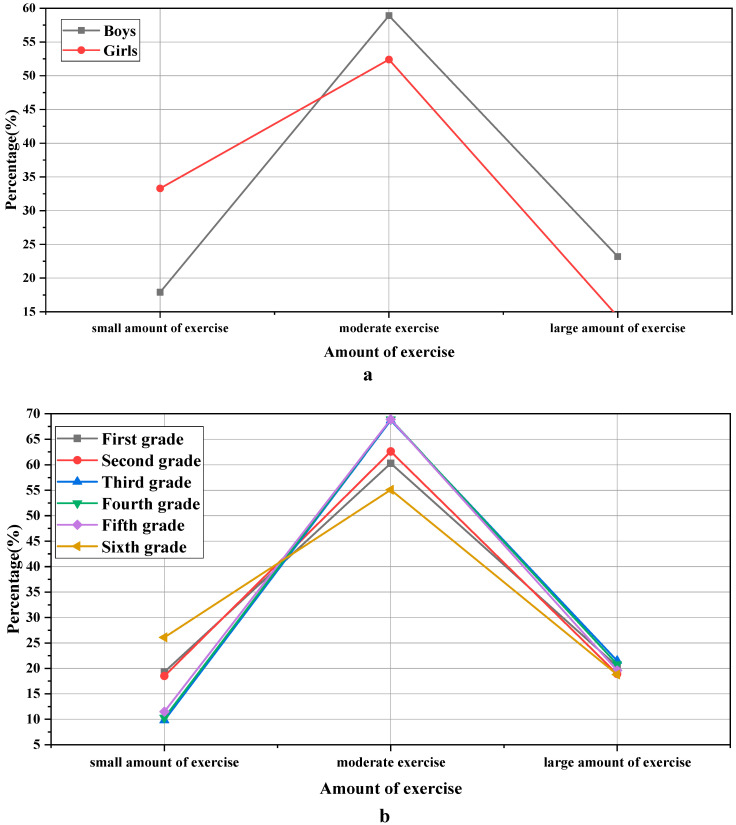
Statistical results of the amount of exercise: (**a**) is the gender classification of the sample; (**b**) is the grade classification of the sample.

**Figure 5 ijerph-19-11451-f005:**
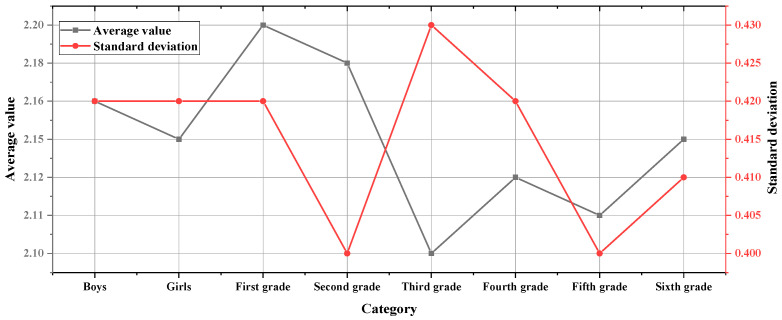
The differences in positive psychological emotions for gender and grade.

**Figure 6 ijerph-19-11451-f006:**
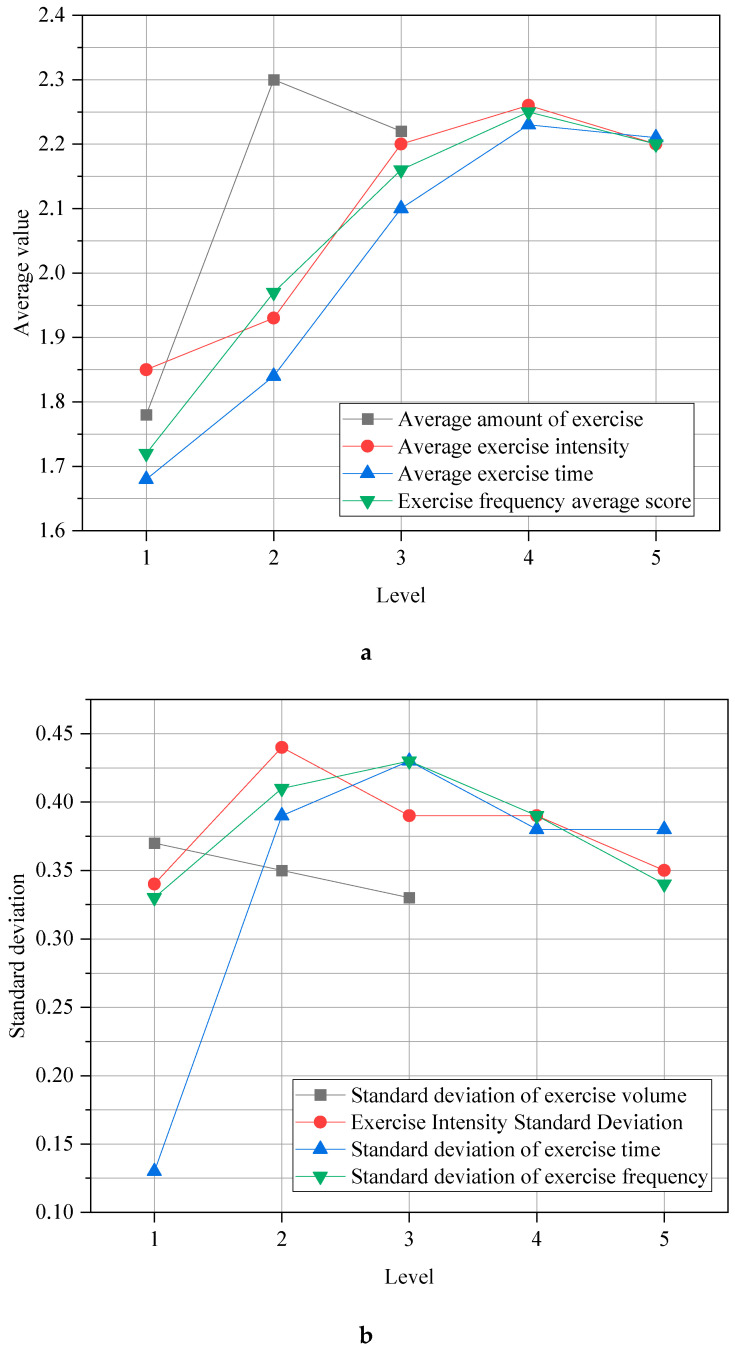
Test results of the difference between positive psychological emotions and physical exercise: (**a**) the mean value of positive emotion in physical exercise; (**b**) the standard deviation of positive emotion in physical exercise.

**Table 1 ijerph-19-11451-t001:** Questionnaire on the physical exercise of primary school students.

Questions	Multiple Choice	
1. How intense is your physical activity? (Test of training intensity)		
A. Light exercise (such as walking, doing radio gymnastics, playing croquet, etc.)		
B. Less intense sports of low intensity (such as recreational volleyball, table tennis, jogging, tai chi, etc.)		
C. Moderate-intensity, more intense, and long-lasting exercise (such as cycling, running, playing table tennis, etc.)		
D. High-intensity, but not long-lasting sports (such as more than half an hour of badminton, basketball, volleyball, tennis, football, etc.)		
E. High-intensity long-lasting exercise with shortness of breath and sweating a lot (such as more than 1 h of running, aerobics, swimming, etc.)		
2. How many minutes do you do the above physical exercise? (Test for training time)		
A. Less than 10 min	B. 11–20 min	C. 21–30 min
D. 31–59 min	E. More than 60 min	
3. How often do you engage in the above physical activity? (Test of training frequency)		
A. Less than once a month	B. 2–3 times a month	
C. 1–2 times a week	D. 3–5 times a week	
E. About once a day		

Parental input during the collection phase of the data on younger children is a limitation of this research and a possible factor undermining the reliability and validity of the findings for younger participants.

**Table 2 ijerph-19-11451-t002:** The positive emotion scale.

The following are some adjectives that describe emotions. It doesn’t matter whether they are good or bad. Please choose the option that best matches your life experience according to your general life experience.					
	Never	Rarely	Sometimes	Often	Always
Interested	□	□	□	□	□
Excited	□	□	□	□	□
Strong	□	□	□	□	□
Passionate	□	□	□	□	□
Proudly	□	□	□	□	□
Alert	□	□	□	□	□
Inspired	□	□	□	□	□
Determined	□	□	□	□	□
Attentive	□	□	□	□	□
Active	□	□	□	□	□

**Table 3 ijerph-19-11451-t003:** Basic information of students.

Grades	Boys	Girls	Total
Grade 1	25	28	53
Grade 2	22	29	51
Grade 3	22	28	50
Grade 4	12	20	32
Grade 5	23	30	53
Grade 6	18	23	41

**Table 4 ijerph-19-11451-t004:** The significance test of positive emotions in physical exercise.

Positive Emotions	Amount of Exercise	Exercise Intensity	Exercise Time	Exercise Frequency
f	197.272 ***	26.518 ***	17.909 ***	20.854 ***

*** stands for <0.001.

**Table 5 ijerph-19-11451-t005:** The results of the correlation analysis between physical exercise and positive psychological emotions.

	Amount of Exercise	Exercise Intensity	Exercise Time	Exercise Frequency
**Positive emotions**	0.403	0.250	0.221	0.238

**Table 6 ijerph-19-11451-t006:** Results of regression analysis of physical exercise on positive emotions.

Calibration Variable	Predictor Variable	B	Beta	t	F	R	R^2^
**positive emotions**	**Constant term**	1.946	-	78.58	92.79 ***	0.297	0.088
	**Physical exercise**	0.007	0.297	9.63			

*** stands for <0.001.

## Data Availability

The raw data supporting the conclusions of this article will be made available by the authors without undue reservation.

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
