# Peer review of "The Effect of Persistence of Physical Exercise on the Positive Psychological Emotions of Primary School Students under the STEAM Education Concept"

_ijerph, 2022, doi:10.3390/ijerph191811451_

Round 1

Reviewer 1 Report

How has the emotion scale been developed? I am missing an explanation of how was this instrument validated.

While with the Physical Activity Rating Scale you provide a reference of the authors that first developed the instrument, as well as the the realiability and validity values, the Positive Emotion Scale is just referenced as including items from D’Souza et al. (2021) and Albert et al. (2021) who applied the instrument to adult population.

My concern is that the Positive Emotion Scale might not be such a reliable and valid instrument as the Physical Activity Rating Scale.

I would like to hear from the authors of this paper if the Positive Emotion Scale had been previously validated with a pilot sample before the present study was conducted.

Author Response

How has the emotion scale been developed? I am missing an explanation of how was this instrument validated.

Reply: Thank you for your suggestions. We have explained the application and development of the positive emotion scale in Section 2.3.

While with the Physical Activity Rating Scale you provide a reference of the authors that first developed the instrument, as well as the the realiability and validity values, the Positive Emotion Scale is just referenced as including items from D’Souza et al. (2021) and Albert et al. (2021) who applied the instrument to adult population.

My concern is that the Positive Emotion Scale might not be such a reliable and valid instrument as the Physical Activity Rating Scale.

Reply: Thank you for pointing out our problems in time. We have tested the reliability and validity of the positive emotion scale in Section 2.3 and supplemented the relevant contents.

I would like to hear from the authors of this paper if the Positive Emotion Scale had been previously validated with a pilot sample before the present study was conducted.

Reply: Thank you very much for your comments on our article. We have supplemented the relevant literature on the application of the positive emotion scale in the introduction, supplemented the reliability and reliability test of the positive emotion scale in the second part, and validated the positive emotion scale.

Reviewer 2 Report

Hello authors
Please find attached my comments on your manuscript for 'The Effect of Persistence of Physical Exercise on the Positive Psychological Emotions of Primary School Students under the STEAM Education Concept'.
Best wishes for your research and future publications.

Author Response

Hello authors
Please find attached my comments on your manuscript for 'The Effect of Persistence of Physical Exercise on the Positive Psychological Emotions of Primary School Students under the STEAM Education Concept'.
Best wishes for your research and future publications.

Dear author(s), Thank you for the opportunity to review this manuscript. You have embarked on a study to provide suggestions for schools and others regarding the effect of persistence of physical exercise on positive psychological emotions for Primary School students. My feedback for your research is as follows:

Reply: Thank you very much for checking our manuscript in your busy schedule.

TITLE

‘The Effect of Persistence of Physical Exercise on the Positive Psychological Emotions of Primary School Students under the STEAM Education Concept.

The title is clear, bound (primary School students) and within an educational context (the STEAM concept).

Please note that the focus of your title needs to resonate throughout the article, that is, the “effect” of persistence of physical exercise on positive psychological emotions. This needs to be clear in the Introduction, Discussion and Conclusion paragraphs.

Reply: Thank you for pointing out our problems in time. We have revised the introduction, discussion, and conclusion to make it more consistent with the main purpose of the article.

ABSTRACT

The abstract clearly outlines the research focus, approach, methods, and findings. Consider

summarising the key results (L23 26) and providing more detail regarding the positive emotions

scale that underpins your positive psychological emotions aspect.

The final sentence is at odds with your manuscript title, with secondary school students mentioned.

Reply: Thank you for pointing out our problems in time. We have revised some contents of the abstract according to the suggestions, and the inconsistency with the title was corrected.

INTRODUCTION

Comments: Is there a research hypothesis? What is this empirical research intending to prove or disprove? A hypothesis would add rigour and clarity to this research, especially when foregrounded against current literature and other studies with a similar research focus.

Reply: Thank you for your comments. There is no research hypothesis in the paper. We have supplemented the research in the first part of the introduction.

It is not clear where this research is situated. Avoid assuming reader knowledge especially pertaining to specific initiatives and programmes in the research country. More information and detail are needed to foreground the research (country, referenced educational concept). The term “persistence”in the title, as a focus of this study also needs to be explained clearly, in context.

You have included a range of studies to provide background to your research. It is important to refer to these literature sources in your discussion section, to synthesise your research results against the findings of other reputable studies.

Reply: Thank you for your comments. We have explained and supplemented the persistence of the title in the introduction, and revised the contents of the discussion according to the suggestions.

Recommendations: Line 43, L48. “At this stage” it is not clear what “stage” is being referred to.

Reply: Thank you for your suggestions. At the beginning of the second paragraph of the introduction, we have revised the "stage".

L50. Sp. demoted

Reply: Thank you very much for your comments on our article. We have revised the relevant contents.

L51. Revise spacing…). Zhu et al. (year)…

Reply: Thank you very much for your comments on our article. We have revised the literature and spacing.

L52. Levels of state anxiety. Clarify what this refers to.

Reply: Thank you for your comments. We have revised the content expression.

L55. Improve the self-efficacy of emotional regulation -not clear.

Reply: Thank you very much for your comments and suggestions. We have revised the wording.

L58-60. Re-write. Awkward construction.

Reply: Thank you for your questions. We have revised the relevant contents in the original L58-60.

L60-64. Move to the start of the introduction paragraph.

Reply: Thank you for your questions. We have moved the expression to the start of the introduction.

L65-75. Avoid repeating exactly what is in the abstract.

Reply: Thank you for your comments. We have revised the contents of the abstract and the third paragraph of the introduction, so that the content is not repeated.

*Use the term’ subjects, or’participants’ rather than ‘objects when referring to the children in the research.

Reply: Thank you for your suggestion. When referring to the children in the study, we have changed "objects" to "participants".

*Remove page numbers from the manuscript unless referring to a direct quote.

Reply: Thank you for your questions. We have deleted the page numbers in the manuscript, except for the directly cited part.

METHODS

Comments: Your methods section includes relevant information about the STEAM educational concept(2.1) which would be better placed in the Introduction, when first mentioned. The Methods section needs to explain and outline the research approach, participant details, and the process by which the data has been collected and analysed, with clear links to the research question (research focus and intention, such as the hypothesis set for this empirical study).

Reply: Thank you for your questions. In the methods section, we have provided supplementary explanations on the overview of research methods and the process of data collection and analysis.

 Additionally, the positive emotions (2.2) paragraph may be better placed in the introduction or in a section after the introduction that provides depth of discussion synthesised from recent research (review of literature). The actual Methods appears to start at 2.3. The research situation (country, province, type of schools), the number of participants per school, the age range, and the selection method need to be included (albeit briefly) to ensure the validity of the collected data. That is, a similar study would generate similar results. I would like to read more detail about the analysis process and other methods you may have used to generate results. I propose that you clearly state your theoretical lens to sharpen the focus of the article. This will also help you to discuss your results in more depth using the theory/concepts you choose.

Reply: Thank you very much for your comments and suggestions. We have revised the title of the second part of the article and the titles of sections 2.1 and 2.2 to make them more consistent with the actual content of the article. We have also revised the steam education theory to highlight the focus of this paper.

There is limited information regarding the participants (children), apart from their grade year. Please outline their demographic to provide greater validity to your method, findings, and discussion sections. I would also like to read about how the Positive Emotions scale was adapted for different age groups, and how children aged five years responded to questions/ a survey regarding the intensity, time, and frequency of their exercise. Were adults or researchers involved in accessing this data? This is a main flaw in the research methodology as questionnaires and surveys must be at the comprehension level of the participants, for collected data to be valid. This may also account for research in this area being mostly on adult participants, due to the subtle nuances of the terms in the emotions scale.

Reply: Thank you for pointing out the problems in the article. At the end of Table 1, we have made supplementary explanations on the filling of the scale and how students complete the questionnaire. We have supplemented the demographic data on participants as shown in Table 3.

Recommendations: L76. Check the use of the term o to ensure there is no confusion with the

research subjects/ participants/ children.

Reply: Thank you for your suggestions. We have checked and revised the title of Part II.

L125. Replace it with the subject being discussed.

Reply: Thank you for your comments. We have revised its contents.

L144. Re-format Table 1. to improve clarity. Acknowledge source. Was the scale adjusted for age, how was it administered?

Reply: Thank you for your comments. We have adjusted Table 1 and supplemented how the survey was carried out.

L145-150. How were the 10 adjectives arrived at for this age group of participants? Did the children understand the subtle difference between never and rarely, or often and always? Were the terms adjusted for different student year groups?

Reply: Thank you very much for your comments and suggestions. For the positive emotion scale, in terms of understanding, parents are required to assist the child in completing it, parents do a good job in assisting, explain it to the students, and students answer the questions with the help of parents. In terms of word understanding, appropriate adjustments have been made for different grades.

L147. “Grades” used to describe student educational levels and Likert Scale choices.

Reply: Thank you very much for pointing out the problem. We have revised the wording of "grades".

L152-157. Limited detail or discussion of the specific positive emotions selected for investigation except as a general term positive emotions.

Reply: Thank you very much for your comments and suggestions. We have added more details of the specific positive emotions selected for investigation.

Suggestions: Quantitative research provides statistical evidence to inform the significance or not of various relational factors. This research provides ample statistical data and analysis on the intensity, time, and frequency of physical exercise however the parameters of what physical exercise is, is vague, therefore it is unclear what is being reported on. More information is needed regarding what ‘intensity’, means relevant to the various year groups under study.

Reply: Thank you for your comments and suggestions. In Section 2.3, we have supplemented the quantification of the physical exercise parameters "intensity, time and frequency".

*Perhaps a reduced age group study would provide greater clarity about the “effects” persistence of physical exercise has on children’s positive emotions.

Reply: Thank you for your comments. We have revised the relevant contents as suggested.

*Remove page numbers from the manuscript unless referring to a direct quote.

Reply: Thank you very much for your comments and suggestions. We have deleted the page numbers in the manuscript, except for the directly cited part.

RESULTS

Comments: This section provides sufficient statistical information to demonstrate rigour in the analysis process. The area of concern is the lack of adjustment to the Positive Emotions Scale for the various Year grades, which undermines the validity and reliability of the figures presented.

Reply: Thank you for your questions. Appropriate adjustments will be made to the expression of the scale for different grades. Students of different grades will have different explanations of variables.

Overall, the author/s have ensured that the data and analysis of the data in this manuscript connects to the purpose/research question(s). The inclusion of the term “effects” would be helpful in this section to aid this link.

Reply: Thank you for your suggestions. We have included the word "effect" in this section.

Take note that the x axis for Figure 3 and Figure 4 are both labelled as Time, whereas Figure 4 is expressing frequency of physical exercise bouts (1-2, 3-5). Does this include no exercise, once a day, or more than once a day?

Reply: Thank you very much for pointing out the problems. Figure 4 shows the statistics according to the third question in Table 1. The statistics are carried out in the time range of week. The last item represents seven times a week, that is, the word "exercise every day".

Recommendations:

L161. Use the term ‘subjects, or ‘participants’ rather than ‘objects when referring to the children in the research.  

Reply: Thank you very much for your comments and suggestions. We have modified it by changing "objects" to "subjects".

L186. Figure 3 means that there is an extremely significant difference between boys and girls in the exercise time of each physical exercise. Add evidence to support this (numerical).

Reply: Thank you very much for pointing out the problem. We have explained the results in Figure 3.

L193.  ‘Improvement’ of grades=graduation from Grade 1 to Grade 6. Replace improvement with “graduation” or a similar term.   

Reply: Thank you for your suggestions. We have modified the 'improvement'.

L201. Participants referred to as male and female previous terminology boys and girls (children).

Reply: Thank you for your comments. We have revised the gender representation of participants.

L231 and L308. Secondary school students included here. This is confusing when the research

specifically states Primary School students.

Reply: Thank you very much for your comments and suggestions. We have revised the expression of this part to change middle school students into primary school students.

L255 -257 “About once a day” and “about once a day” -is this correct?

Reply: Thank you for your comments. It should be "about once a day", not "about once".

L284. ‘…relevant rationalization suggestions?’ What dos this mean?

Reply: Thank you for your suggestions. We have revised its expression.

Suggestions: *Balance the presentation of physical exercise data and positive emotions data equally.

*Combine the eight exercise line graphs into two or three, without losing relevant information.

Reply: Thank you for your comments. We have adjusted the figure, as shown in Figure 7.

*Remove the 3D images (Figure 1), instead present sample data in table form.

Reply: Thank you very much for your comments and suggestions. We have modified Figure 1 into a table, as shown in Table 3.

DISCUSSION AND SUGGESTIONS

L275-285. The suggestion of one 30-minute plus bout of physical activity per week is not supported by other international research, recommendations, and guidelines for children aged 5 10 years. For example, children under 12-years-of age are recommended to take part on 60-minutes of moderate to vigorous physical activity per day, five days per week (NZMOH, SportNZ).

Reply: Thank you for your comments. We have modified this part.

L152 157. Limited detail or discussion of the specific positive emotions selected for investigation

except as a general term-positive emotions.

Reply: Thank you very much for your comments and suggestions. We have revised the details discussed in this part.

Recommendations: L286-295. Limited or no connection between this paragraph and the results

presented, as results and/or discussion of the positive emotions scale is limited.

Reply: Thank you very much for pointing out the problem. We have revised this part to make it more consistent with the theme of the article.

L296-306. This paragraph focuses on the benefits of sport. This type of physical exercise needs to be flagged in the abstract, introduction, and review of literature if the authors consider this to be a valid recommendation.

Reply: Thank you for your suggestions. We have marked this part in the abstract, introduction, and literature review.

*Discussion section needs to align this research to the results and findings of other current research (literature review section) on the effects of regular moderate to vigorous exercise on positive emotions and menta wellbeing.

Reply: Thank you for your comments and suggestions. We have revised some contents of the discussion according to the suggestions.

*Include suggestions and/or recommendations as part of the conclusion paragraph.

Reply: Thank you for your suggestions. We have included the contents of the proposed department as part of the conclusion.

CONCLUSION

This paragraph is long and reiterates data presented in the Results section. The purpose of the

conclusion is to conclude the article by highlighting key findings, acknowledging research limitations, and making suggestions and recommendations for future developments (considering the findings) and to address the limitations of the research design, and/or implementation. Include recommendations or suggestions for improvement, further research, changes to methodology, data analysis, or research focus, thereby identifying further opportunities for research ongoing, to build on the outcomes of this study.

Reply: Thank you for your comments and suggestions. We have adjusted the conclusion part.

Suggestions: *When making a recommendation from your research findings, start with a summary of the result that predicates the suggested improvement. For example, “This research also proves the effect of physical exercise on positive emotions (insert reference to a table or figure/s here). Therefore, schools are advised to …

Reply: Thank you for your comments. We have revised the conclusions according to the recommendations.

* Avoid making blanket statements or generalizations unless backed up with relevant literature and/or empirical evidence. L289-290…. “Students should be able to cheer themselves up in the face of difficulties, change the unfavourable conditions in reality, and improve their positive emotions in the face of setbacks.”

Reply: Thank you for your comments. We have revised and adjusted this part according to the suggestions.

REFERENCES and FORMATTING

The overall quality of writing in this manuscript was good, however, some terms have been used incorrectly, or may create confusion due to the nuances of the term, for example, the use of object/subject. Specifically, the Methodology and Discussion sections lack rigour. The presentation of figures needs to be addressed. Regarding formatting, all tables and figures need to “fit” within the document margins, and page numbers need to be removed from the manuscript unless acknowledging a direct quote by the author/s. The tools used for collecting data (PARS-3, Positive Emotions Scale) are credible and appropriate but must be referenced fully. Neither appears in the reference list.

Reply: Thank you very much for your comments and suggestions. We have revised the methods and contents of the article as required. The format was adjusted, and the page numbers in the manuscript were deleted. The chart format has been adjusted so that all tables and charts "match" the document margins.

Similarly, the statistical data analysis tools (Cronbach’s coefficient, SPSS software tool) are appropriate and credible for this research however, neither are listed in the references section.

Suggestions: Most of the supporting literature focuses on adult studies and specific health or mental health issues for adults. The authors may be better served to source research and literature pertaining specifically to children aged 5-10 years, to ensure the supporting studies are relevant to the participant demographic being researched.

Reply: Thank you very much for your suggestion. We have revised some references according to the recommendations. To ensure that the supporting study is demographically relevant to the participants being studied.

Round 2

Reviewer 2 Report

Thank you authors for your revised manuscript.

The abstract is much improved with greater clarity and increased relevant information. 

*Note these suggested amendments: 

L23-24 Change 'and the score of boys is higher than that of girls' TO 'with the boys scoring higher than the girls'

L31. Change 'and the correlation coefficient is 0.297' TO 'with a correlation coefficient of 0.297'.

L34-34. Change 'and it has played a certain role in strengthening the physical exercise and mental health of primary and secondary school students' TO 'and has played a role in strengthening the physical exercise and mental health of primary and secondary school students.' 

L34-35. Change 'and it is recommended to pay attention to physical exercise of primary school students' TO 'Greater attention to the physical exercise of primary school students is recommended'. 

The introduction 

* Note these suggested amendments 

L59.  Replace or remove 'momentous' unless you provide evidence to support this adjective.

L62. Remove the page number as this appears to be incorrect (pp. 100408). Page numbers are required for direct quotes only.

L65-66. See note for L62.

L73-74. Change from 'the psychological and emotional effects of primary school students' TO 'the psychological and emotional effects of persistence of physical exercise on primary school students'.

L75-76. Change 'sports disciplines, STEAM education integrates the characteristics of various disciplines, and is a fusion and creation of diverse disciplines' TO 'sports disciplines. STEAM education integrates the characteristics of various disciplines, and is a fusion and creation of diverse disciplines. (two sentences)

L76-79. Remove 'To improve the comprehensive quality of the current Chinese primary school students, this research analyses the effect of the persistence of physical exercise on the psychological and emotional effects of primary school students under the STEAM education concept'. This sentence is repetitive, without providing new information.

L81. Change ' first,' TO ' firstly,'

L82. Change 'Then, ' TO 'Secondly,' ...

L84. Change '. And through this analysis....' TO 'Through this analysis ...'

L86. Change ' At last,' TO ' Finally, '

L87. Change 'relevant rationalization suggestions are made' TO 'relevant suggestions are made.'

L87. Change 'This research has a certain' TO 'This research aims to provide ...'

Methods 

* Note these suggested amendments

L102-106. Change this long sentence into two more concise sentences, TO 'It is in line with the research direction of China's teaching reform to focus on improving the teaching mode and integration of teaching objectives, through various thinking modes. These reforms aim to strengthen the attractiveness of physical exercise to students, and to promote students' self-motivation [10].'

L107. Spelling 'STAEM' TO 'STEAM'.

L108-109. Change ', learn to listen to the opinions of others, and learn to try a variety of ideas to solve problems.' TO ', such as, learn to listen to the opinions of others and to try a variety of ideas to solve problems.'

L120. Change 'It can add' TO 'Positive emotions can add ...'

L127-129. Change 'that it can increase the adrenaline in the blood, the hormone that mobilizes the power of the organism, thereby making the struggler more empowered to achieve his goals [18,19].' TO  'that positive emotions can increase adrenaline in the blood, the hormone that mobilises the power of the organism, thereby making the person more empowered to achieve their goals [18,19].'

L141. Change 'The scoring method is written in equation' TO 'The scoring method is written as an equation' ....

L143-144. Review the literature source for 'Exercise intensity and exercise frequency are divided into 1-5 cases, with 1-5 points, respectively, and exercise time is split into 1-5 types, with 0-4 points, respectively [23].' This would be clearer in table form.

L145-146. Refer to Table 1 for the following 'it is split into five intensities from A to E (Table 1.). Repeat for exercise frequency and exercise time.

Table 1 is much improved. The explanation of procedures for data collection is also improved with important detail regarding parental help to manage the various age groups and comprehension levels of the participants. 

NB. The input of parents in the data collection stage for lower grade children must be highlighted as a limitation for this research, and a possible factor in undermining the reliability and validity of the findings for lower grade participants.

Results and Analysis

L186. Change 'scales'. Are you referring to a questionnaire, a survey or something else? What does 'scales' refer to? This term has not been used previously.

Figure 1.b. Label the x axis the same as Figure 1.a. - 'intensity level'

L258-259. Change 'of primary and schoolchildren' TO 'of primary schoolchildren'

L275-276. This table needs to be labelled as Table 2 and referred to within your manuscript.  

L293-294. Change 'Figure 7.' TO Table 3. Refer to Table 3 in your manuscript.

L302-303. As above. Change the figure to Table 4. Refer to Table 4 in your manuscript.

L310. Change 'And the best time' TO 'The best time ...'

L316. Change 'also hopes to improve5' TO 'also aims to improve'.

L318. Change 'should expand the popularization of' TO 'are recommended to popularise mental health knowledge, ....'

L320. Change 'and regarding physical exercise' TO 'and to regard physical exercise...'

Conclusion

L323. Remove 's' from heading.

L324-325. Change this statement as the research specifically relates to Primary aged children. The results of surveying 5-10-year-old children cannot be extrapolated to 11-18-year-old students. Include a recommendation to extend the research to a similar sized secondary school cohort for a future study.

L324-326. Change TO 'The purpose was to promote improvement of all-around comprehensive quality of primary education and strengthen the emphasis on physical exercise and mental health levels of primary school students.' 

The conclusion still contains results, findings and some analysis. Include a Discussion section for L332-338 using past tense throughout. RELATE to literature presented in the introduction.

The conclusion needs to include L324 - 328 and L349 - 358.

L356-358. Change 'This research has a certain positive effect on promoting the comprehensive development of primary and secondary school students.' TO 'This research had a positive effect on promoting the comprehensive development of primary school students.'

Limitations

Key limitations of the study need to be acknowledged with recommendations to address these limitations in future research.

Author Response

Thank you authors for your revised manuscript.

The abstract is much improved with greater clarity and increased relevant information. 

*Note these suggested amendments: 

L23-24 Change 'and the score of boys is higher than that of girls' TO 'with the boys scoring higher than the girls'

Reply: Thank you for your suggestion. According to your suggestion, 'and the score of boys is higher than that of girls' has been changed to 'with the boys scoring higher than the girls'

L31. Change 'and the correlation coefficient is 0.297' TO 'with a correlation coefficient of 0.297'.

Reply: Thanks for your comment. According to your comment, 'and the correlation coefficient is 0.297' has been changed to 'with a correlation coefficient of 0.297'

L34-34. Change 'and it has played a certain role in strengthening the physical exercise and mental health of primary and secondary school students' TO 'and has played a role in strengthening the physical exercise and mental health of primary and secondary school students.' 

Reply: Thank you for your reminder. According to your reminder, 'and it has played a certain role in strengthening the physical exercise and mental health of primary and secondary school students' has been changed to 'and has played a role in strengthening the physical exercise and mental health of primary and secondary school students.' 

L34-35. Change 'and it is recommended to pay attention to physical exercise of primary school students' TO 'Greater attention to the physical exercise of primary school students is recommended'. 

Reply: Thanks for your guidance. According to your guidance, 'and it is recommended to pay attention to physical exercise of primary school students' has been changed to 'Greater attention to the physical exercise of primary school students is recommended'.

The introduction 

* Note these suggested amendments 

L59.  Replace or remove 'momentous' unless you provide evidence to support this adjective.

Reply: Thank you for your suggestion. Based on your suggestion, 'momentous' has been removed.

L62. Remove the page number as this appears to be incorrect (pp. 100408). Page numbers are required for direct quotes only.

Reply: Thanks for your comment. Based on your comment, the page number (pp. 100408) has been removed.

L65-66. See note for L62.

Reply: Thank you for your reminder. Based on your reminder, we have adjusted with reference to L62's comments.

L73-74. Change from 'the psychological and emotional effects of primary school students' TO 'the psychological and emotional effects of persistence of physical exercise on primary school students'.

Reply: Thanks for your guidance. According to your guidance, 'the psychological and emotional effects of primary school students' has been changed to 'the psychological and emotional effects of persistence of physical exercise on primary school students'.

L75-76. Change 'sports disciplines, STEAM education integrates the characteristics of various disciplines, and is a fusion and creation of diverse disciplines' TO 'sports disciplines. STEAM education integrates the characteristics of various disciplines, and is a fusion and creation of diverse disciplines. (two sentences)

Reply: Thank you for your suggestion. According to your suggesting, 'sports disciplines, STEAM education integrates the characteristics of various disciplines, and is a fusion and creation of diverse disciplines' has been changed to 'sports disciplines. STEAM education integrates the characteristics of various disciplines, and is a fusion and creation of diverse disciplines.

L76-79. Remove 'To improve the comprehensive quality of the current Chinese primary school students, this research analyses the effect of the persistence of physical exercise on the psychological and emotional effects of primary school students under the STEAM education concept'. This sentence is repetitive, without providing new information.

Reply: Thanks for your comment. Based on your comment, 'To improve the comprehensive quality of the current Chinese primary school students, this research analyses the effect of the persistence of physical exercise on the psychological and emotional effects of primary school students under the STEAM education concept' has been removed.

L81. Change ' first,' TO ' firstly,'

Reply: Thank you for your reminder. According to your reminder, ' first,' has been changed to ' firstly,'

L82. Change 'Then, ' TO 'Secondly,' ...

Reply: Thanks for your guidance. According to your guidance, ' Then,' has been changed to ' Secondly,'

L84. Change '. And through this analysis....' TO 'Through this analysis ...'

Reply: Thank you for your suggestion. According to your suggestion, 'And through this analysis....' has been changed to 'Through this analysis ...'

L86. Change ' At last,' TO ' Finally, '

Reply: Thanks for your comment. According to your suggestion, ' At last,' has been changed to ' Finally, '

L87. Change 'relevant rationalization suggestions are made' TO 'relevant suggestions are made.'

Reply: Thank you for your reminder. According to your reminder, 'relevant rationalization suggestions are made' has been changed to 'relevant suggestions are made.'

L87. Change 'This research has a certain' TO 'This research aims to provide ...'

Reply: Thank you for your suggestion. According to your suggestion, 'This research has a certain' has been changed to 'This research aims to provide ...'

Methods 

* Note these suggested amendments

L102-106. Change this long sentence into two more concise sentences, TO 'It is in line with the research direction of China's teaching reform to focus on improving the teaching mode and integration of teaching objectives, through various thinking modes. These reforms aim to strengthen the attractiveness of physical exercise to students, and to promote students' self-motivation [10].'

Reply: Thank you for your suggestion. According to your suggestion, this long sentence into two more concise sentences, TO 'It is in line with the research direction of China's teaching reform to focus on improving the teaching mode and integration of teaching objectives, through various thinking modes. These reforms aim to strengthen the attractiveness of physical exercise to students, and to promote students' self-motivation [10].'

L107. Spelling 'STAEM' TO 'STEAM'.

Reply: Thanks for your comment. According to your comment, 'STAEM' has been changed to 'STEAM'.

L108-109. Change ', learn to listen to the opinions of others, and learn to try a variety of ideas to solve problems.' TO ', such as, learn to listen to the opinions of others and to try a variety of ideas to solve problems.'

Reply: Thank you for your reminder. According to your reminder, 'learn to listen to the opinions of others, and learn to try a variety of ideas to solve problems.' has been changed to ', such as, learn to listen to the opinions of others and to try a variety of ideas to solve problems.'

L120. Change 'It can add' TO 'Positive emotions can add ...'

Reply: Thanks for your careful reading. According to your suggestion, 'It can add' has been changed to 'Positive emotions can add ...'

L127-129. Change 'that it can increase the adrenaline in the blood, the hormone that mobilizes the power of the organism, thereby making the struggler more empowered to achieve his goals [18,19].' TO  'that positive emotions can increase adrenaline in the blood, the hormone that mobilises the power of the organism, thereby making the person more empowered to achieve their goals [18,19].'

Reply: Thanks for your guidance. According to your guidance, 'that it can increase the adrenaline in the blood, the hormone that mobilizes the power of the organism, thereby making the struggler more empowered to achieve his goals [18,19].' has been changed to 'that positive emotions can increase adrenaline in the blood, the hormone that mobilises the power of the organism, thereby making the person more empowered to achieve their goals [18,19].'

L141. Change 'The scoring method is written in equation' TO 'The scoring method is written as an equation' ....

Reply: Thank you for your suggestion. According to your suggestion, 'The scoring method is written in equation' has been changed to 'The scoring method is written as an equation'

L143-144. Review the literature source for 'Exercise intensity and exercise frequency are divided into 1-5 cases, with 1-5 points, respectively, and exercise time is split into 1-5 types, with 0-4 points, respectively [23].' This would be clearer in table form.

Reply: Thanks for your comment. References have been provided for this section based on your comments.

L145-146. Refer to Table 1 for the following 'it is split into five intensities from A to E (Table 1.). Repeat for exercise frequency and exercise time.

Reply: Thank you for your reminder. Remarks on exercise time and exercise frequency have been added to Table 1 based on your reminder, improving the value of Table 1.

Table 1 is much improved. The explanation of procedures for data collection is also improved with important detail regarding parental help to manage the various age groups and comprehension levels of the participants. 

  1. The input of parents in the data collection stage for lower grade children must be highlighted as a limitation for this research, and a possible factor in undermining the reliability and validity of the findings for lower grade participants.

Reply: Thanks for your guidance. According to your guidance, Parental input in the data collection phase for younger children has been highlighted below Table 1 as a limitation of this study and a possible factor undermining the reliability and validity of the findings for younger participants.

Results and Analysis

L186. Change 'scales'. Are you referring to a questionnaire, a survey or something else? What does 'scales' refer to? This term has not been used previously.

Reply: Thank you for your suggestion. 'scales' has been modified according to your suggestion.

Figure 1.b. Label the x axis the same as Figure 1.a. - 'intensity level'

Reply: Thanks for your comment. The abscissa of Figure 1b has been modified to 'intensity level' based on your comment.

L258-259. Change 'of primary and schoolchildren' TO 'of primary schoolchildren'

Reply: Thank you for your reminder. According to your reminder, 'of primary and schoolchildren' has been changed to 'of primary schoolchildren'.

L275-276. This table needs to be labelled as Table 2 and referred to within your manuscript.  

Reply: Thanks for your guidance. Figure 6c has been modified to Table 4 according to your guidance.

L293-294. Change 'Figure 7.' TO Table 3. Refer to Table 3 in your manuscript.

Reply: Thanks for your careful reading. Figure 7 has been modified to Table 5 according to your suggestion.

L302-303. As above. Change the figure to Table 4. Refer to Table 4 in your manuscript.

Reply: Thank you for your suggestion. Figure 8 has been modified to Table 6 according to your suggestion.

L310. Change 'And the best time' TO 'The best time ...'

Reply: Thanks for your comment. According to your comment, 'And the best time' has been changed to 'The best time ...'

L316. Change 'also hopes to improve5' TO 'also aims to improve'.

Reply: Thank you for your reminder. According to your reminder, 'also hopes to improve5' has been changed to 'also aims to improve'.

L318. Change 'should expand the popularization of' TO 'are recommended to popularise mental health knowledge, ....'

Reply: Thanks for your guidance. According to your guidance, 'should expand the popularization of' has been changed to 'are recommended to popularise mental health knowledge, ....'

L320. Change 'and regarding physical exercise' TO 'and to regard physical exercise...'

Reply: Thank you for your suggestion. According to your suggestion, 'and regarding physical exercise' has been changed to 'and to regard physical exercise...'

Conclusion

L323. Remove 's' from heading.

Reply: Thank you for your suggestion. The 's' in the title has been removed as per your suggestion.

L324-325. Change this statement as the research specifically relates to Primary aged children. The results of surveying 5-10-year-old children cannot be extrapolated to 11-18-year-old students. Include a recommendation to extend the research to a similar sized secondary school cohort for a future study.

Reply: Thanks for your comment. The Conclusion section has been revised based on your comments, thereby narrowing the groups covered by the findings of this paper.

L324-326. Change TO 'The purpose was to promote improvement of all-around comprehensive quality of primary education and strengthen the emphasis on physical exercise and mental health levels of primary school students.' 

Reply: Thank you for your reminder. According to your reminder, the first sentence of the conclusion section has been changed to 'The purpose was to promote improvement of all-around comprehensive quality of primary education and strengthen the emphasis on physical exercise and mental health levels of primary school students.'

The conclusion still contains results, findings and some analysis. Include a Discussion section for L332-338 using past tense throughout. RELATE to literature presented in the introduction.

Reply: Thanks for your guidance. The syntax of the conclusion section has been modified according to your guidance, thereby improving the plausibility of the content of the conclusion.

The conclusion needs to include L324 - 328 and L349 - 358.

Reply: Thanks for your careful reading. The content of the conclusion section has been revised based on your suggestion, increasing its value.

L356-358. Change 'This research has a certain positive effect on promoting the comprehensive development of primary and secondary school students.' TO 'This research had a positive effect on promoting the comprehensive development of primary school students.'

Reply: Thank you for your suggestion. According to your suggestion, 'This research has a certain positive effect on promoting the comprehensive development of primary and secondary school students.' has been changed to 'This research had a positive effect on promoting the comprehensive development of primary school students.'

Limitations

Key limitations of the study need to be acknowledged with recommendations to address these limitations in future research.

Reply: Thanks for your comment. Based on your conclusions, a Limitations section has been added to the text, where a discussion of the limitations of this research has been added, along with corresponding suggestions for future research.